# (-)-Fenchone Prevents Cysteamine-Induced Duodenal Ulcers and Accelerates Healing Promoting Re-Epithelialization of Gastric Ulcers in Rats via Antioxidant and Immunomodulatory Mechanisms

**DOI:** 10.3390/ph17050641

**Published:** 2024-05-15

**Authors:** Maria Elaine Cristina Araruna, Edvaldo Balbino Alves Júnior, Catarina Alves de Lima Serafim, Matheus Marley Bezerra Pessoa, Michelle Liz de Souza Pessôa, Vitória Pereira Alves, Marcelo Sobral da Silva, Marianna Vieira Sobral, Adriano Francisco Alves, Mayara Karla dos Santos Nunes, Aurigena Antunes Araújo, Leônia Maria Batista

**Affiliations:** 1Postgraduate Program in Natural and Synthetic Bioactive Products, Health Sciences Center, Federal University of Paraiba (UFPB), João Pessoa 58050-585, PB, Brazil; elaine.araruna@gmail.com (M.E.C.A.); edvaldojunioralves@gmail.com (E.B.A.J.); catarinaalvesdelima@gmail.com (C.A.d.L.S.); mmarleybp@gmail.com (M.M.B.P.); michelleliz2008@hotmail.com (M.L.d.S.P.); pereiravitoria689@gmail.com (V.P.A.); marcelosobral.ufpb@gmail.com (M.S.d.S.); mariannavbs@gmail.com (M.V.S.); 2Department of Pharmaceutical Sciences, IPeFarM, Federal University of Paraiba, João Pessoa 58051-970, PB, Brazil; 3Department of Physiology and Pathology, Health Sciences Center, Federal University of Paraiba (UFPB), João Pessoa 58050-585, PB, Brazil; adriano.alves@academico.ufpb.br (A.F.A.); mayarakarlasn@hotmail.com (M.K.d.S.N.); 4Department of Morphology, Histology and Basic Pathology, Biosciences Center, Federal University of Rio Grande do Norte, Natal 59078-970, RN, Brazil; auriprinino@gmail.com

**Keywords:** (-)-Fenchone, gastric ulcer, duodenal ulcer, healing, monoterpene, immunomodulatory

## Abstract

Background: (-)-Fenchone is a naturally occurring monoterpene found in the essential oils of *Foeniculum vulgare* Mill., *Thuja occidentalis* L., and *Peumus boldus* Molina. Pharmacological studies have reported its antinociceptive, antimicrobial, anti-inflammatory, antidiarrheal, and antioxidant activities. Methods: The preventive antiulcer effects of (-)-Fenchone were assessed through oral pretreatment in cysteamine-induced duodenal lesion models. Gastric healing, the underlying mechanisms, and toxicity after repeated doses were evaluated using the acetic acid-induced gastric ulcer rat model with oral treatment administered for 14 days. Results: In the cysteamine-induced duodenal ulcer model, fenchone (37.5–300 mg/kg) significantly decreased the ulcer area and prevented lesion formation. In the acetic acid-induced ulcer model, fenchone (150 mg/kg) reduced (*p* < 0.001) ulcerative injury. These effects were associated with increased levels of reduced glutathione (GSH), superoxide dismutase (SOD), interleukin (IL)-10, and transforming growth factor-beta (TGF-β). Furthermore, treatment with (-)-Fenchone (150 mg/kg) significantly reduced (*p* < 0.001) malondialdehyde (MDA), myeloperoxidase (MPO), interleukin-1 beta (IL-1β), tumor necrosis factor-alpha (TNF-α), and nuclear transcription factor kappa B (NF-κB). A 14-day oral toxicity investigation revealed no alterations in heart, liver, spleen, or kidney weight, nor in the biochemical and hematological parameters assessed. (-)-Fenchone protected animals from body weight loss while maintaining feed and water intake. Conclusion: (-)-Fenchone exhibits low toxicity, prevents duodenal ulcers, and enhances gastric healing activities. Antioxidant and immunomodulatory properties appear to be involved in its therapeutic effects.

## 1. Introduction

Peptic ulcers are inflammatory or necrotizing lesions that can extend from the mucosal layer to deeper layers like the submucosa, muscle, and serosa, and are often found in the esophagus, stomach, or duodenum [1,2,3]. This condition, arising from an imbalance between protective and aggressive factors, poses a significant global health concern, being the most prevalent gastrointestinal ailment, affecting millions worldwide, with high recurrence rates [3,4,5].

Healing constitutes a vital response to tissue injury, entailing a complex and sequential process involving systemic components, such as hemostasis, inflammatory cells, chemokines, cytokines, matrix molecules, and nutrients, with an elevated metabolic demand. These phases, which include inflammation, proliferation, and remodeling, occur concurrently [6].

Current therapeutic approaches for peptic ulcers typically focus on inhibiting gastric acid secretion or enhancing mucosal protective factors [7,8]. However, challenges such as drug interactions and long-term side effects often limit their efficacy, prompting the exploration of alternative therapies involving natural products [7,9].

In recent years, various plant species and their isolated compounds, notably terpenes, have demonstrated promising effects in peptic ulcer studies employing animal models [10,11]. Terpenes, constituting around 90% of essential oil constituents [12], find widespread use in industries, for instance, in fragrances, agrochemicals, and nutraceuticals [13].

(-)-Fenchone (1,3,3-trimethyl bicyclo [2.2.1] heptane-2-one), a bicyclic monoterpene derived from two isoprene units and featuring an organic ketone function, is present in essential oils of various plants like *Foeniculum vulgare* Mill. [14], *Thuja occidentalis* L. [15], and *Peumus boldus* Molina [16]. This phytoconstituent exhibits diverse pharmacological activities, including antinociceptive [17], antimicrobial [18], anti-inflammatory, antidiarrheal [19], and antioxidant [20,21] properties.

Building upon these promising findings, this study aimed to assess the gastric healing potential of (-)-Fenchone using the acetic acid induction model, evaluate its duodenal antiulcer activity induced by cysteamine, investigate oral toxicity in repeated doses over 14 days, and elucidate the mechanisms of action underlying its effects.

## 2. Results

### 2.1. Effect of (-)-Fenchone Oral Pretreatment on Duodenal Ulcer Formation

Results from the cysteamine-induced duodenal ulcer induction model in rats demonstrated that oral administration of lansoprazole (30 mg/kg) or (-)-fenchone at doses of 37.5, 75, 150, and 300 mg/kg significantly decreased ULA (*p* < 0.001) compared to the negative control group. The inhibition percentages of injury were 60%, 31%, 68%, 74%, and 76%, respectively (see Figure 1 and Figure 2).

#### Histological Analysis of Duodenal Ulcer Formation

Histological sections of duodena stained with hematoxylin and eosin from the normal, lansoprazole, and (-)-Fenchone groups (Figure 3A,C,D) revealed intact villi (V) and enterocytes without cytostructural alterations. Conversely, in the negative control group, extensive ulceration of the epithelium, loss of epithelial cells, transmural inflammatory exudate with serosal involvement, and perforation were evident. Masson’s trichrome staining showed that (-)-Fenchone (Figure 3G) significantly reduced extracellular matrix deposition (*p* < 0.05, 9.00 [5,6,7,8,9,10,11,12]) compared to the negative control (*p* < 0.05, 8.00 [6,7,8,9,10,11]) (Figure 3F).

### 2.2. Effect of (-)-Fenchone on Gastric Healing after 14 Days of Oral Treatment

The results for the acetic acid-induced gastric ulcer model in rats showed that (-)-fenchone (150 mg/kg) and lansoprazole (30 mg/kg) significantly reduced ULA (*p* < 0.001) to 10.15 ± 2.3 and 16.48 ± 2.5 mm^2^ when compared to the negative control with 32.60 ± 3.2 mm^2^, presenting percentages of gastric healing of 69 and 49%, respectively (Figure 4).

#### 2.2.1. Histological Analysis of Gastric Healing

In the histological sections of the stomach stained with hematoxylin and eosin (Figure 5A–D), the stomach mucosae (M) are visible. In animals from the sham (A), lansoprazole-treated (C), and (-)-Fenchone-treated (D) groups, the integrity of the mucosae and gastric glands is evident. However, in contrast to these groups, animals in the negative control group exhibit severe mucosal destruction and ulcer formation, characterized by organ surface necrosis, and intense acute inflammatory reactions (*).

In the histological sections stained with Masson’s trichrome (Figure 5E–H) for the sham (E) group 12.00 (10–19), the lansoprazole-treated (G) group 15.00 (11–19), and the (-)-Fenchone-treated (H) group 15.00 (10–21), the integrity of the stomach mucosae (M) and gastric glands was observed. However, in animals from the negative control group (*p* < 0.001) 45.00 (31–57) (F), an abundance of scar tissue (∞) was evident in the area of the mucosal ulcer.

#### 2.2.2. Effect of (-)-Fenchone on the Modulation of Antioxidant and Anti-Inflammatory Properties during Gastric Healing

##### GSH

Referring to the antioxidant activity of (-)-Fenchone, the group of animals pretreated with the vehicle alone (5% tween 80) showed a reduction in GSH levels to 67.2 ± 6.5 nmol of GSH/mg protein compared to the sham group (90.9 ± 2.8 nmol GSH/mg protein). However, previous administration of lansoprazole (30 mg/kg) (90.0 ± 6.2 nmol of GSH/mg of proteins) or (-)-Fenchone (150 mg/kg) (79.9 ± 9.4 nmol GSH/mg protein) restored baseline GSH levels (Figure 6A).

##### SOD

Animals in the negative control group demonstrated a significant reduction in the activity of the SOD enzyme to 3.5 ± 0.86 U of SOD/mg of protein compared to the sham group (7.5 ± 0.76 U SOD/mg protein). However, SOD activity was significantly increased (*p* < 0.001) with the previous administration of lansoprazole (30 mg/kg) or (-)-Fenchone (150 mg/kg) to 6.2 ± 0.52 and 5.9 ± 0.48 U of SOD/mg of protein, respectively, when compared to the negative control (Figure 6B).

##### MDA

The negative control group showed an increase in MDA levels to 78.5 ± 8.0 nmol MDA/g tissue compared to the sham group (42.9 ± 5.8 nmol MDA/g tissue). However, the administration of lansoprazole (30 mg/kg) (63.5 ± 3.3 nmol of MDA/g tissue) or (-)-Fenchone (150 mg/kg) (57.9 ± 6.4 nmol MDA/g of tissue) restored MDA levels to baseline conditions (Figure 6C).

##### MPO

The results showed that in the negative control group, there was an increase in MPO levels to 28.1 ± 4.0 units of MPO/g of tissue when compared to the sham group (8.5 ± 1.4 units MPO/g tissue). However, the groups treated with lansoprazole (30 mg/kg) or (-)-Fenchone (150 mg/kg) showed MPO levels significantly reduced to 7.3 ± 2.5 and 5.7 ± 1.7 units of MPO/g of tissue in comparison with the control group, respectively (Figure 6D).

##### Levels of IL-1β, TNF-α, and IL-10 Cytokines

In the evaluation of immunomodulatory effects by measuring cytokines, (-)-Fenchone significantly decreased (*p* < 0.001) the levels of IL-1β (pg/mL) to 258. ± 24.58 compared to the negative control (423.6 ± 26.30) (Figure 7A). TNF-α levels (pg/mL) were also decreased after treatment with (-)-Fenchone (1398 ± 86.20) when compared to the negative control (2069 ± 83.80) (Figure 7B). Furthermore, (-)-Fenchone prevented (*p* < 0.001) a reduction in IL-10 (pg/mL) (207.4 ± 17.30) in comparison to the negative control (94.67 ± 15.54) (Figure 7C).

##### Immunohistochemical Analysis for NF-kB and TGF-β in Gastric Samples Submitted to Acetic Acid-Induced Gastric Ulcer Model

Stomachs treated with (-)-Fenchone following exposure to acetic acid displayed unevenly distributed and focal NF-kB positivity (brown color) akin to the sham group. Immunostaining was significantly reduced to 36.0 (30.0–46.0) μm^2^ compared to the negative control group, in which the staining measured 59.0 (48.0–80.0) μm^2^ and was particularly noticeable in the ulcer area (Figure 8). Regarding immunostaining for TGF-β, (-)-Fenchone treatment resulted in a significant increase in positively marked cells for TGF-β, measuring 204.0 (201.0–221.0) μm^2^ compared to the negative control group, in which staining measured 104.0 (100.0–107.0) μm^2^ (Figure 9).

#### 2.2.3. Effect of (-)-Fenchone on Low Toxicity after Repeated Doses

After delineating the pharmacological effects, the safety of the phytoconstituent was evaluated following 14 consecutive oral administrations once a day. The results revealed that animals treated with (-)-Fenchone (150 mg/kg) exhibited a significant increase (*p* < 0.01) in water intake (ml) to 243.9 ± 11.3 compared to the group treated with 5% Tween 80 (207.6 ± 21.7). Additionally, the treatment led to an increase in feed intake (g) to 193 ± 14.8 and prevented (*p* < 0.001) body weight loss (42.33 ± 2.0) compared to the negative control group (35.57 ± 2.6) (Table 1).

In the macroscopic assessment of the heart, liver, spleen, and kidneys, no alterations were observed between the sham, control, and treated groups. Moreover, no significant changes (*p* > 0.05) were noted in the organ index across all analyzed groups (Table 2). Additionally, there were no significant differences (*p* > 0.05) in glucose, urea, creatinine, alkaline phosphatase, AST, and ALT levels compared to the respective control groups (Table 2).

## 3. Discussion

Duodenal ulcer stands as the most prevalent form of peptic ulcer affecting populations worldwide. Its pathogenesis is intertwined with escalated aggressive factors, such as acid-peptic secretion and proteases, coupled with diminished mucosal protective elements and potential *H. pylori* infection [22,23]. The experimental induction of duodenal ulcers in rats via cysteamine has been extensively utilized to emulate this pathological state [23,24].

Cysteamine (β-mercaptoethylamine) swiftly induces ulceration, escalates gastric acid secretion, diminishes somatostatin bioavailability [25], and elevates serum gastrin levels [26], thereby fostering increased gastric acid production and impeding defensive mechanisms like alkaline mucus secretion and bicarbonate levels in the proximal duodenum, ultimately culminating in duodenal ulcer formation [24]. Renowned for inducing rapid ulceration, cysteamine stands out as one of the prime agents for simulating acute and chronic duodenal ulcers in animal models [25]. Its administration induces oxidative stress by generating H_2_O_2_, which, in the presence of metallic ions like Fe^3+^, engenders highly reactive species within duodenal enterocytes [22]. Studies have demonstrated that cysteamine undermines antioxidant defenses [24,27,28], leading to alterations in the redox state and a decline in duodenal mucosal oxygenation, evident through dysregulated GPx and augmented release of duodenal endothelin-1, a potent vasoconstrictor, thereby compromising mucosal blood perfusion and resulting in ischemia and hypoxia [23,28].

(-)-Fenchone, at the assessed doses (37.5, 75, 150, and 300 mg/kg), significantly mitigated duodenal ulcerative lesions compared to the negative controls. A study by Carvalho et al. (2014) with the monoterpene geraniol (7.5 mg/kg) showcased a protective effect on duodenal mucosae [29]. Furthermore, these findings align with those observed for other monoterpenes, such as β-myrcene (7.5 mg/kg) [30] and p-cymene (25, 50, 100, and 200 mg/kg) [31], demonstrating antiulcerogenic activity in the same evaluated model.

These outcomes were further elucidated through histomorphological analyses utilizing hematoxylin and eosin (HE) staining, wherein a reduction in cysteamine-induced duodenal tissue lesions and inflammatory exudates at the lesion site was apparent. Masson’s trichrome staining revealed escalated extracellular matrix deposition in lesions within the negative control group (5% Tween 80), while the (-)-Fenchone-treated group exhibited a reduction in such deposition. Previous studies with monoterpenes have unveiled similar protective effects on duodenal mucosae, exemplified by Carvalho et al. (2014) with geraniol [29] and Bonamin et al. (2014) with β-myrcene, in the same model [30], attributed to reduced MPO levels and heightened GSH levels.

Hence, the duodenal antiulcer activity demonstrated in this study may be attributed to antioxidant effects, reinforcement of mucosal protective factors, and a reduction in inflammatory cell infiltration. Nevertheless, further experiments are warranted to elucidate the precise mechanism underlying the protective action of (-)-Fenchone against cysteamine-induced duodenal lesions.

Chronicity exacerbates gastric ulcers, leading to episodes of remission and recurrence [24]. The model induced by acetic acid closely mimics human ulcers in terms of location, severity, chronicity, and healing process, thus making it widely utilized in evaluating agents with potential pro-healing effects on chronic gastrointestinal lesions [24,32]. The most efficacious dose of (-)-Fenchone (150 mg/kg) identified in the duodenal ulcer evaluation was selected for assessing gastric healing and toxicity following repeated doses.

In this regimen, treatment with (-)-Fenchone (150 mg/kg) for 14 days augmented the healing percentage by 69% (reducing ALU [*p* < 0.001] to 10.15 ± 2.3) compared to the negative control group. Similar effects have been documented in the literature for the monoterpene (-)-myrtenol (50 mg/kg) [33], thymol (30 and 100 mg/kg) [34], geraniol (3 mg/kg) [10], and *p*-cymene [31].

Histomorphological analysis of slides stained with HE revealed that (-)-Fenchone treatment preserved tissue architecture and gastric glands, accompanied by reduced inflammatory infiltrate. Masson’s trichrome staining illustrated the integrity of stomach mucosae and gastric glands in the (-)-Fenchone group, distinct from the control group exhibiting increased collagen fiber deposition and abundant scar tissue. Studies reported in the literature with monoterpenes further support these findings [31,35,36].

In this context, the data presented herein offer compelling evidence of the healing properties exhibited by (-)-Fenchone. Subsequently, investigations into the mechanisms of healing were conducted utilizing gastric tissue samples from an acetic acid-induced gastric ulcer model.

The excessive generation of reactive oxygen species within gastric tissue disrupts cell membrane integrity and permeability, culminating in cell death and facilitating ulcer formation [36,37]. Thus, the evaluation of antioxidant activity involved assessing the levels of GSH (mg of non-protein sulfhydryl groups per g of tissue), SOD (units per g of protein), MDA (nmol per g of tissue) in gastric samples, as well as MPO (units per g of tissue). The antioxidant system operates within gastric mucosal cells to mitigate or prevent cytotoxicity induced by oxidative stress, employing enzymes like SOD and compounds such as GSH [38].

Administration of (-)-Fenchone over 14 days demonstrated an antioxidant effect, evidenced by elevated levels of GSH and SOD, alongside reduced levels of MDA and MPO. Consequently, the therapeutic efficacy observed in our study may stem from the inhibition of free radicals, leading to a subsequent decrease in toxic lipoperoxides, or from the promotion of endogenous antioxidant synthesis, thereby maintaining cellular redox equilibrium.

Monoterpenes such as 1,8-cineole [35,36], thymol [34], and limonene [39] have been documented to enhance antioxidant defenses. These findings corroborate the antioxidant properties observed in our investigation, suggesting that the attenuation of ROS activity by the substance under scrutiny potentially underlies the reduction in acetic acid-induced injury extent, fostering a conducive milieu for the healing process.

Gastric healing represents a multifaceted process encompassing inflammation, cell migration, proliferation, re-epithelialization, angiogenesis, tissue remodeling, and extracellular matrix (ECM) deposition. These intricacies are orchestrated by growth factors and transcription factors activated in response to tissue injury, operating in a concerted manner [40,41]. Considering the complexity, therapeutic compounds exhibiting increased growth factors and anti-inflammatory activity emerge as promising candidates for the prevention and treatment of human peptic ulcers [33,42].

Within the injured gastric mucosa, the inflammatory cascade is triggered, marked by an upsurge in mediators such as NF-kB, which amplifies the expression of inflammatory responses, leading to heightened release of IL-1, IL-6, and TNF-α, among other factors [43,44]. Excessive IL-1β expression exacerbates ulcer formation and fosters TNF-α production [45], resulting in heightened levels of IL-1 and TNF linked to neutrophil infiltration and epithelial cell apoptosis. These events diminish gastric microcirculation around the ulcer site, impeding ulcer healing progress [44].

IL-10 acts as a protective cytokine within the gastric mucosa, signaling an anti-inflammatory response that dampens the immune reaction mediated by inflammatory cells. This effect manifests through the inhibition of TNF-α feedback, subsequently curbing the inflammatory cascade in gastric tissue [46].

Treatment with (-)-Fenchone attenuated the levels of inflammatory cytokines (TNF-α and IL-1β) while maintaining IL-10 cytokine levels close to baseline, suggestive of an immunomodulatory effect. Monoterpenes such as limonene [39] and carvacrol [47] have demonstrated similar immunomodulatory effects, aligning with our observed outcomes.

The results found in the present study demonstrate that treatment by repeated doses for 14 days with (-)-Fenchone reduced immunostaining for NF-kB when compared to the negative control groups. The NF-kB pathway plays a key role in the regulation of genes related to an acute inflammatory reaction. After exposure to various inflammatory stimuli, NF-kB is activated, translocated to the nucleus, and causes transcriptional activation of other pro-inflammatory cytokines [48,49]. The healing activity of this substance seems to be, at least in part due to the downregulation of the NF-kB pathway, a key player in pro-inflammatory cytokine production. Furthermore, treatment with (-)-Fenchone increased the immunostaining of TGF-B, a growth factor involved in the migration of fibroblasts and myofibroblasts, stimulation of angiogenesis, and collagenization in gastric tissues [50]. These findings suggest an immunomodulatory activity of (-)-Fenchone, owing to its ability to maintain basal levels of anti-inflammatory cytokines while reducing levels of inflammatory cytokines.

These results indicate that the substance under investigation stimulates endogenous healing mechanisms by inducing multiple factors crucial for gastric ulcer healing, including cell proliferation and angiogenesis. These findings align with a study conducted with the monoterpenes d-limonene and p-cymene in acetic acid-induced ulcers, wherein treatments exhibited healing effects attributed to increased growth factors promoting gastric lesion re-epithelialization [31,51].

Furthermore, in this model of acetic acid-induced gastric ulcers, we assessed the safety of (-)-Fenchone treatment following 14 consecutive oral administrations once daily. Data from the supplier of the substance, Sigma Aldrich, indicate low acute toxicity of (-)-Fenchone with an LD50 Oral in female rats of 2000 mg/kg (OECD Test Guideline 423), with no observed mortality at this dose. Seeking to validate previous studies and enhance toxicological understanding, toxicity was evaluated using the most effective dose of (-)-Fenchone (150 mg/kg). No changes in body mass were noted throughout the treatment period, and organ weights were unaffected. Additionally, biochemical parameter evaluation indicated no detrimental effects on hepatic function (AST and ALT) or renal function (urea and creatinine). (-)-Fenchone treatment at a dose of 150 mg/kg daily for 14 days did not alter any of the biochemical or hematologic parameters analyzed. These findings suggest that oral treatment with (-)-Fenchone for 14 days at a dosage of 150 mg/kg daily does not induce subacute toxicity in rats.

In summary, the results demonstrate that (-)-Fenchone, a compound originally derived from natural products, not only prevents duodenal ulcers but also exhibits gastric ulcer healing effects. The enhancement in gastric healing activity appears to be attributed to mechanisms involving a reduction in oxidative stress (increased GSH and SOD with decreased MDA and MPO) and immunomodulation (reduced TNF-α, IL-1β, and NF-kB and increased TGF-β and IL-10 levels). These results are complemented by the safety profile observed during the 14-day treatment period.

## 4. Materials and Methods

### 4.1. Animals

Male Wistar rats (Rattus norvegicus) weighing between 180 and 250 g were obtained from the Animal Production Unit (APU) of the Institute for Research on Drugs and Medicines at the Federal University of Paraiba (IPeFarM/UFPB). These rats were acclimated to the local vivarium conditions, maintained at a temperature of 23 ± 2 °C with a 12 h light–dark cycle, and provided with ad libitum access to industrial pellet food and water. All experimental procedures adhered to international principles for laboratory animal research [52] and received approval from the Institutional Ethics Commission on Animal Use (CEUA/UFPB) under registration number 7216040119/19. Euthanasia was performed using an anesthetic overdose administered via intra-peritoneal injection, consisting of xylazine (2%, 10 mg/kg) and ketamine (5%, 180 mg/kg), following internationally accepted guidelines for laboratory animal care. Every effort was made to minimize the number of animals used and alleviate their pain, suffering, and stress.

### 4.2. Reagents

The following substances were used in this study: (-)-Fenchone (SIGMA Chemical Co., St. Louis, MO, USA) solubilized in 5% Tween 80; sodium acetate (SIGMA Chemical Co., St. Louis, MO, USA); acetonitrile (SIGMA Chemical Co., St. Louis, MO, USA); 5,5′-ditiobis-2-nitrobenzoic acid (DTNB) (SIGMA Chemical Co., St. Louis, MO, USA); ethylenediaminetetraacetic acid (EDTA) (SIGMA Chemical Co., St. Louis, MO, USA); glacial acetic acid (SIGMA Chemical Co., St. Louis, MO, USA); trichloroacetic acid (SIGMA Chemical Co., St. Louis, MO, USA); antirat antibodies for IL-1β, TNF-α, and IL-10 (R&D systems); biotinylated sheep polyclonal antibodies (anti-IL-1β, anti-TNF-α, or anti-IL-10) (R&D Systems); immunohistochemical polyclonal rat antibodies (NFκB or TGFβ) (Cloud-clone Corp); sodium carbonate (MERK, Darmstadt, DE-HE, Germany); potassium chloride (SIGMA Chemical Co., USA); magnesium chloride (SIGMA Chemical Co., St. Louis, MO, USA); Cysteamine hydrochloride (SIGMA Chemical Co., St. Louis, MO, USA); sodium chloride PA (QUIMEX-MERCK, Uberaba, MG, Brazil); ethyl ether (MERK, Darmstadt, DE-HE, Germany); monobasic sodium phosphate (SIGMA Chemical Co., St. Louis, MO, USA); bibasic sodium phosphate (SIGMA Chemical Co., St. Louis, MO, USA); lansoprazole (SIGMA Chemical Co., St. Louis, MO, USA); Methyl-Phenylindole (SIGMA Chemical Co., St. Louis, MO, USA); ketamine 5% (VETANARCOL); Tris Buffer (Vetec^®^); Trizma Buffer (SIGMA Chemical Co., St. Louis, MO, USA); and xylazine 2% (DORCIPEC).

### 4.3. Preventive Antiulcer Activity Evaluation on Cysteamine-Induced Duodenal Ulcer Model

Rats (N = 6–7) underwent a 12 h fasting period and were then divided into groups: a normal group (no treatment) and groups treated orally (10 mL/kg) with lansoprazole 30 mg/kg (standard control), 5% Tween 80 (negative control), and (-)-Fenchone at doses of 37.5, 75, 150, and 300 mg/kg. Following a 1 h pretreatment period, cysteamine hydrochloride (300 mg/kg) was administered orally twice, with a 4 h interval between doses. After 24 h, the animals were euthanized, and their small intestines were removed and opened along the antimesenteric region. The intestines were then placed between glass plates to facilitate imaging using a digital camera for lesion determination. Quantification of ulcer areas (UA, mm^2^) was performed using AVSoft Bioview^®^ software version 4.0.1.

### 4.4. Healing Assessment on Acetic Acid-Induced Gastric Ulcer Model

Wistar rats (N = 8–10), following a 12 h fasting period, were randomly divided into six groups: sham (healthy group), standard control (treated with lansoprazole 30 mg/kg), negative control (administered 5% Tween 80), and (-)-Fenchone groups (150 mg/kg). The optimal dose of (-)-Fenchone was determined based on its efficacy in the cysteamine-induced duodenal ulcer model, selected according to the highest lesion inhibition percentages among the evaluated doses. The animals were anesthetized with ketamine and xylazine before undergoing laparotomy. The stomachs were exposed, and the outer serosal surfaces were treated with 100 µL of acetic acid applied via a cotton pellet confined within a 5.0 mm circular diameter plastic tube for 1 min. Subsequently, the stomachs were rinsed with deionized water, and the abdominal walls were sutured with cotton thread. Following 48 h of gastric lesion induction, the animals were orally treated (10 mL/kg) once daily for 14 days, following the previously described regimen. This treatment protocol was administered to all groups except the sham group, which only underwent laparotomy and immediate suturing [53]. On the fifteenth day, the animals were euthanized, and their stomachs were excised and opened along the greater curvature for the determination of the ulcerative lesion area (ULA) using a digital caliper. Additionally, the healing percentage (%) was calculated as described below.
$$ULA({mm}^{2})=(ulcer\;length)\times (ulcer\;width)$$
$$Healing(\%)=\frac{\left(Negative\;control\;ULA-Treated\;group\;ULA\right)}{Negative\;control\;ULA}\times 100$$

Following ULA determination, tissue samples encompassing gastric injuries were collected, stored, and preserved in a 10% buffered formaldehyde solution for subsequent histological and immunohistochemical analysis. Tissues designated for biochemical assays were dissected, weighed, and then frozen at −80 °C.

#### 4.4.1. Histological and Morphometric Analysis

Fragments of gastric tissue from the acetic acid-induced gastric ulcer protocol and duodenal tissue from the cysteamine-induced ulcer protocol were preserved in a 10% buffered formaldehyde solution until histological processing. The tissues were then embedded in histological paraffin and sectioned to a thickness of 4 µm. Following sectioning, two stains were applied: hematoxylin and eosin (HE) and Masson’s trichrome (MT). Histological sections stained with Masson’s trichrome were visualized using the 40× objective of an Olympus microscope (Tokyo, Japan). Twenty random images were captured and digitized using the same microscope and Q-Color3 microscope software. In each image, all pixels exhibiting shades of blue (indicative of Masson’s trichrome staining) were selected to generate a binary image. These binary images were digitally processed to calculate the area in μm^2^.

#### 4.4.2. Determination of GSH, MDA, MPO, and SOD

##### GSH Determination

The GSH levels were determined following the protocol established by Faure and Lafond (1995) [54]. Tissue samples were suspended in 0.02 M EDTA at a 1:10 (*v*/*v*) ratio. Subsequently, 400 µL of this homogenate was extracted, to which 320 µL of distilled water and 80 µL of 50% trichloroacetic acid were added. The mixture was then centrifuged at 3000 rpm at 4 °C for 15 min. Following centrifugation, 100 µL of the resulting supernatant was transferred to a 96-well microplate, along with 200 µL of Tris and 25 µL of DTNB. The microplate was incubated at room temperature, and after 15 min, spectrophotometric readings were taken at a wavelength of 412 nm using a Polaris spectrophotometer. A calibration curve was constructed using reduced L-glutathione. GSH values for the samples were determined by interpolating the values against the standard curve and expressed in nmol GSH/mg of protein.

##### MDA Determination

The tissue samples were suspended in Trizma^®^ buffer (Tris HCl) at a ratio of 1:5 (*w*/*v*), homogenized, and then centrifuged at 11,000 rpm at 4 °C for 10 min. Following centrifugation, 300 μL of the supernatants was transferred to Eppendorf tubes, to which 750 μL of the chromogenic compound (10.3 mM of 1-methyl-2-phenylindol) and 225 μL of hydrochloric acid (37%) were added. The mixture was incubated at 45 °C in a water bath for 40 min, followed by another centrifugation at 11,000 rpm at 4 °C for 5 min. Subsequently, 300 μL of the supernatant was transferred to a 96-well microplate, and the absorbance was measured by colorimetry at 586 nm using a plate reader (Polaris’ spectrophotometer). The data were interpolated with the standard curve, and the results were expressed as nmol MDA/g tissue [55].

##### MPO Determination

The tissue sample fragments were homogenized in hexadecyltrimethylammonium bromide (HTAB) buffer, which acts as a detergent, lysing the neutrophil granules containing myeloperoxidase. Following homogenization, the material underwent sonication for 5 min. A double freezing and thawing process was then performed to aid in the disruption of cellular structures and subsequent enzyme release. The homogenate was centrifuged at 5000 rpm at 4 °C for 20 min and concentrated for 24 h. The following day, 7 μL of the supernatant was collected and mixed with 200 μL of the reading solution (o-dianisidine hydrochloride, potassium phosphate buffer, and 1% H_2_O_2_). The absorbance was measured using a spectrophotometer at a wavelength of 450 nm at 0 and 1 min. The results were expressed as units of myeloperoxidase per gram of tissue [56].

##### SOD Determination

Tissue samples were homogenized in phosphate buffer (0.4 M, pH 7.0) and then centrifuged for 15 min at 10,000 rpm at 4 °C. The supernatant was collected and used in the assay. Plates containing the reaction medium (10 mM phosphate buffer), L-methionine (1.79 mg/mL, pH 7.8), riboflavin (0.2 mg/mL, pH 7.8), NBT (1.5 mg/mL, pH 7.8), and 10 μL of the sample supernatant were exposed to a 15 W fluorescent lamp for 10 min. After this incubation period, the material was measured at 630 nm using a spectrophotometer [57].

#### 4.4.3. Immunomodulatory Activity

##### Cytokine Determination

The levels of pro-inflammatory cytokines (IL-1β and TNF-α) and immunoregulatory cytokine (IL-10) were assessed using an enzyme-linked immunosorbent assay (ELISA) of the sandwich type. Captured antibodies specific to each interleukin were immobilized in a 96-well microplate with a flat bottom. Following an 18 h incubation period, the plate was washed with a 0.05% Tween 20 solution (wash buffer), then blocked with a 1% bovine serum albumin solution, and washed again with the wash buffer. Subsequently, tissue macerate was prepared in phosphate-buffered saline (PBS) at a ratio of 100 mg of tissue to 600 μL of PBS, homogenized, and centrifuged at 4000 rpm for 10 min at 4 °C. Supernatants (100 μL) were pipetted onto the 96-well plate to create the standard curve. A biotinylated secondary antibody (100 μL) was then added, followed by a 2 h incubation period and three washes. The plate was further incubated with streptavidin for 20 min, washed three times, and then the substrate for development (DuoSet Kit©—R&D Systems Catalog—DY999) was added and incubated for an additional 20 min. After incubation, the reaction was halted by adding 50 μL of stop solution, and the absorbance was measured at 450 nm using a spectrophotometer. Results were obtained by interpolation from the standard curve and expressed in pg/mL [58].

#### 4.4.4. Immunohistochemical Analysis

Stomach sections (5 μm thick) from 5 animals per group were obtained using a microtome and then transferred onto silanized slides (Dako, Glostrup, Hovedstaden, Denmark). These sections underwent dewaxing and hydration procedures. Subsequently, a simple indirect blocking method was employed using Anti Peroxidase Peroxidase (APP), wherein the primary antibody targeted the antigen (protein) to be detected, while the secondary antibody facilitated binding to the APP complex. The slides were then incubated overnight at 4 °C with primary antibodies against NF-kB and TGF-β. Following thorough washing with distilled water, the slides were treated with a secondary antibody for 60 min. 3,3′-diaminobenzidine (DAB, Biocare Medical, Pacheco, CA, USA) was utilized as the chromogen, and the specimens were counterstained with hematoxylin. The samples were examined under an optical microscope (Olympus microscope, Tokyo, Japan) equipped with a camera (Nikon DS-Ri2).

### 4.5. Evaluation of Toxicity after Repeated Doses

In the acetic acid-induced ulcer model, various parameters, including water and feed intake, as well as the body weight of the animals, were assessed over 14 days. On the 15th day, the animals underwent a 12 h fasting period before being anesthetized for arterial puncture to collect blood samples. Biochemical analyses were conducted on the blood samples collected in gel separator tubes, which were then centrifuged at 1000× *g* for 10 min. Measurements of urea, creatinine, alkaline phosphatase (ALP), aspartate aminotransferase (AST), alanine aminotransferase (ALT), glucose, cholesterol, and triglycerides were performed using animal serum. Hematological analyses were carried out on blood samples collected in EDTA-containing tubes, and parameters such as red blood cell count, hematocrit, hemoglobin level, mean corpuscular volume (MCV), mean corpuscular hemoglobin (MCH), mean corpuscular hemoglobin concentration (MCHC), and global and differential leukocyte counts were determined. Additionally, blood smears were obtained and stained to confirm and control differential leukocyte counts (including neutrophils, lymphocytes, monocytes, and eosinophils) and were analyzed under optical microscopy. Following blood collection, the animals were euthanized, and their organs (heart, liver, spleen, and kidneys) were harvested for macroscopic examination and weight assessment. The analyses were conducted using specific kits for automatic biochemical and hematological analyzers.

### 4.6. Statistical Analysis

Data were expressed as means ± standard deviations (SDs) or means ± standard errors (SEs) for parametric values or expressed as medians (minimum values–maximum values) for non-parametric data. One-way ANOVA (parametric data) or the Kruskal–Wallis test (non-parametric data) was performed, followed by Dunnett’s or Tukey’s or Dunn’s post hoc tests, respectively. The results were considered significant when *p* < 0.05. All data were analyzed using version 5.0 for Windows, GraphPad 5.0 Software, San Diego, California USA.

## 5. Conclusions

Based on the analysis of the results obtained for (-)-Fenchone in models simulating gastric and duodenal ulcers, it can be inferred that fenchone demonstrates low toxicity even with repeated doses over 14 days, as evidenced in the acetic acid-induced gastric ulcer model. Fenchone exhibits notable gastric healing activity, seemingly mediated by antioxidant mechanisms, such as the restoration of GSH and SOD levels, in addition to reductions in MDA and MPO levels. Moreover, its action appears to involve immunomodulatory mechanisms, as indicated by the decrease in IL-1β and TNF-α levels, coupled with an increase in IL-10 levels. Fenchone also displays anti-duodenal ulcer activity, suggesting antioxidant properties akin to those observed in the gastric model, characterized by the restoration of GSH and SOD levels and reductions in MDA and MPO levels. Consequently, it is proposed that fenchone holds promise for further exploration in the development of gastrointestinal ulcer treatments.

## Figures and Tables

**Figure 1 pharmaceuticals-17-00641-f001:**
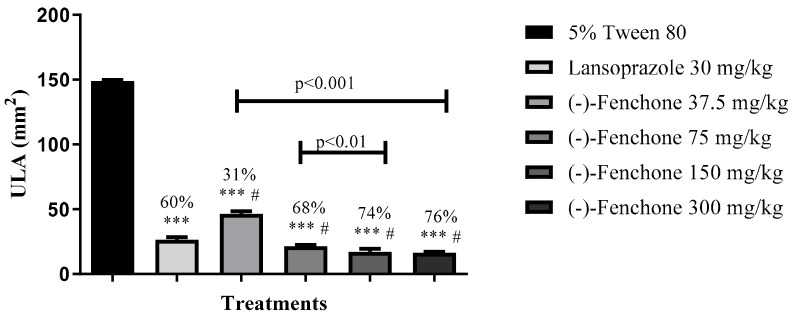
Effect of oral administration of (-)-fenchone on cysteamine-induced duodenal ulcers. Results are expressed as mean ± SD. One-way analysis of variance (ANOVA) was used: F (5.27) = 4.780, followed by Dunnett’s or Tukey’s test performed using the GraphPad 5.0 software. *** *p* < 0.001 compared to the negative control group (0.9% saline solution); # *p* < 0.001 compared to the lansoprazole group (30 mg/kg) (n = 5–7). ULA = ulcerative lesion area.

**Figure 2 pharmaceuticals-17-00641-f002:**
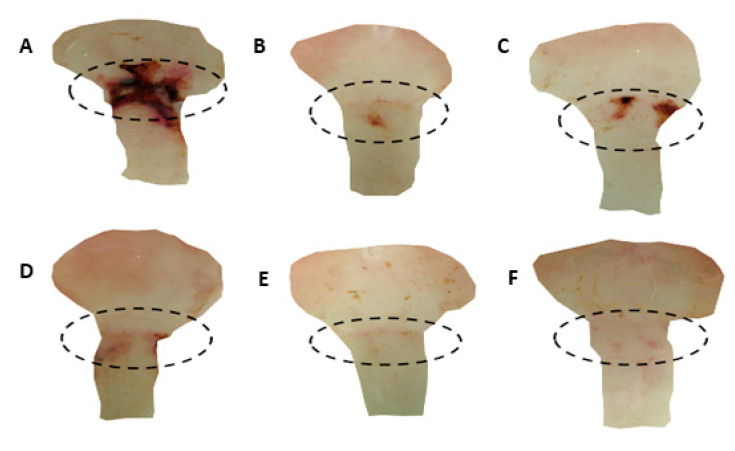
Macroscopic aspects of the duodenal mucosa of male Wistar rats pretreated (p.o.) with 5% tween 80 (**A**), lansoprazole 30 mg/kg (**B**), (-)-Fenchone 37.5 mg/kg (**C**), 75 mg/kg (**D**), 150 mg/kg (**E**), and 300 mg/kg (**F**) in the cysteamine-induced duodenal ulcer model. The dashed line represents the duodenal area of the animals.

**Figure 3 pharmaceuticals-17-00641-f003:**
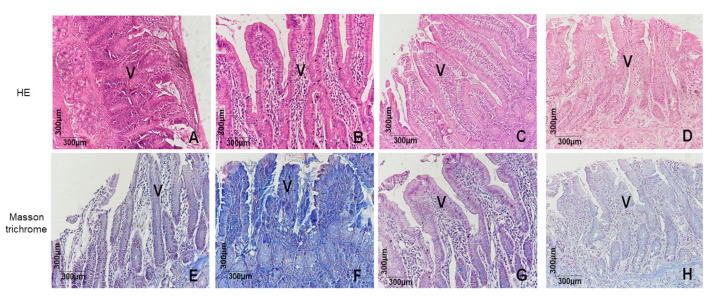
Microscopic examination of the duodenal mucosae of rats subjected to cysteamine-induced duodenal ulcers and treated or untreated with (-)-Fenchone was conducted. Representative photomicrographs of the animals’ duodena from the experimental groups are presented: normal (**A**,**E**), 5% Tween 80 (**B**,**F**), lansoprazole 30 mg/kg (**C**,**G**), and (-)-Fenchone 150 mg/kg (**D**,**H**). Staining with hematoxylin and eosin (HE) (**A**–**D**) and Masson’s trichrome (**E**–**H**) was performed. Intestinal villi and enterocytes are denoted as (V). Graphic morphometric analysis using Masson’s trichrome staining was conducted. Results are expressed as the median (minimum–maximum) of the parameters analyzed (n = 5, three sessions per animal). Statistical analyses were performed using the Kruskal–Wallis and Dunn’s post hoc tests with, GraphPad 5.0 Software. ** *p* < 0.01 or * *p* < 0.05 compared to the control group (5% Tween 80).

**Figure 4 pharmaceuticals-17-00641-f004:**
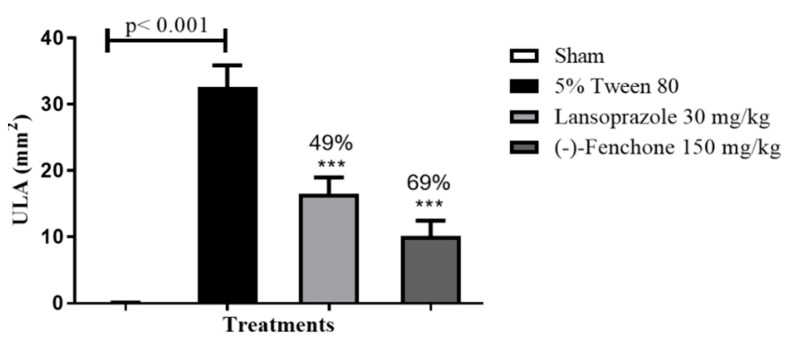
Effect of oral administration of (-)-Fenchone and lansoprazole on acetic acid-induced gastric ulcers. Results are expressed as mean ± SD. One-way ANOVA was used: F (3.23) = 211.6, followed by Dunnett’s or Tukey’s test performed using the GraphPad 5.0 software. *** *p* < 0.001 compared to the negative control group (Tween 80 5%) (n = 6–8). ULA = ulcerative lesion area.

**Figure 5 pharmaceuticals-17-00641-f005:**
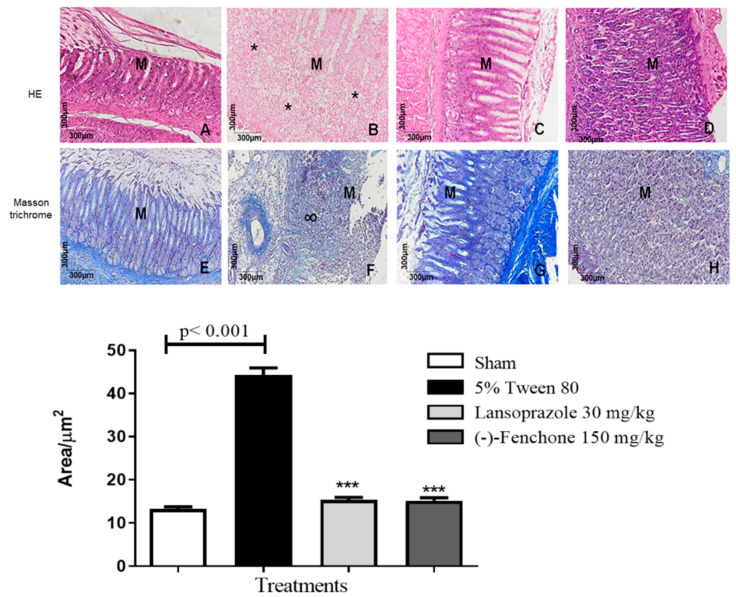
Microscopic effects on the gastric mucosae of rats submitted to acetic acid-induced gastric ulcers and treated or not with (-)-Fenchone. Representative photomicrographs of the animals’ stomachs from the experimental groups: sham (**A**,**E**), 5% tween 80 (**B**,**F**), lansoprazole 30 mg/kg (**C**,**G**), and (-)-Fenchone 150 mg/kg (**D**,**H**). HE staining (**A**–**D**) and Masson’s trichrome staining (**E**–**H**). Stomach mucosa (M), acute inflammatory reaction (*), scarring tissue (∞). Graphic morphometric analysis by Masson’s trichrome staining. Results are expressed as the median (minimum–maximum) of the parameters analyzed (n = 5, three sessions per animal). The Kruskal–Wallis test and Dunn’s posterior test were performed using the Graph Pad Prism 5.0 software. *** *p* < 0.001 vs. negative control group (5% tween 80).

**Figure 6 pharmaceuticals-17-00641-f006:**
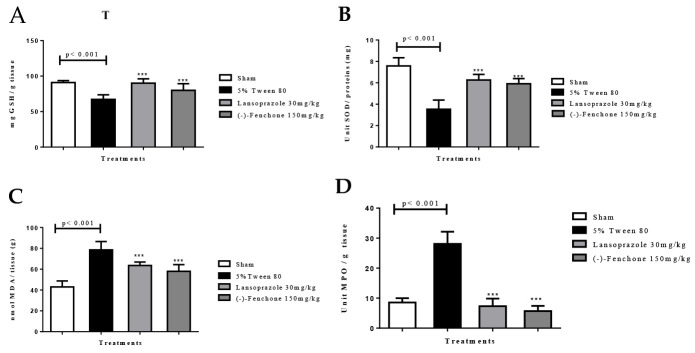
Effect of oral administration of (-)-fenchone and lansoprazole on (**A**) GSH, (**B**) SOD, (**C**) MDA, and (**D**) MPO levels in the gastric ulcer model induced by acetic acid in rats. Results are expressed as the mean ± SE of the parameters analyzed (n = 5–8). One-way ANOVA was followed by Dunnett’s test. *** *p* < 0.001 vs. control group.

**Figure 7 pharmaceuticals-17-00641-f007:**
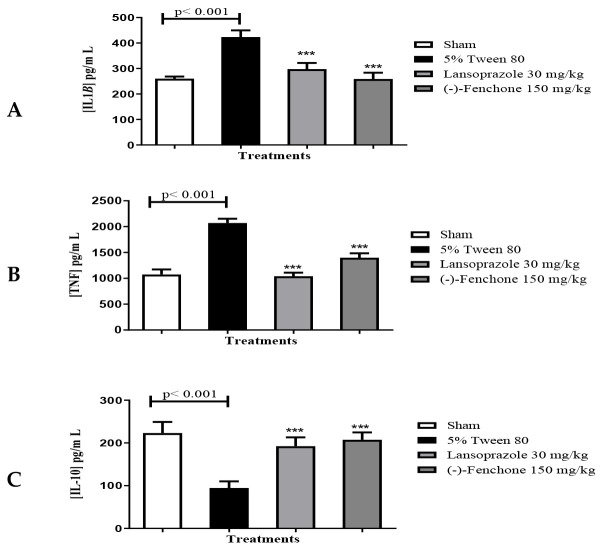
Effects of (-)-fenchone and lansoprazole on (**A**) IL-1β, (**B**) TNF-α, and (**C**) IL-10 levels in the gastric ulcer model induced by acetic acid in rats. Results are expressed as the mean ± SE of the parameters analyzed (n  =  6–8). One-way ANOVA was followed by Dunnett’s or Tukey’s test performed using the GraphPad 5.0 software. *** *p* < 0.001 vs. control group.

**Figure 8 pharmaceuticals-17-00641-f008:**
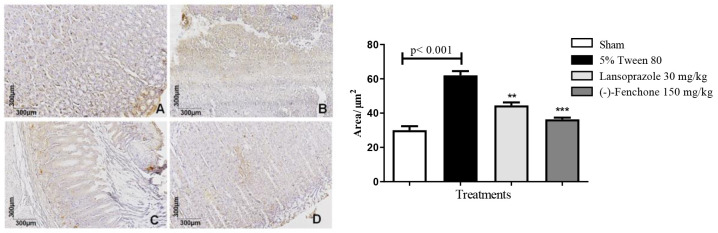
Photomicrographs with immunohistochemical marking for NF-kB in stomach samples from rats: (**A**) sham, (**B**) 5% Tween 80 (10 mL/kg), (**C**) Lansoprazole (30 mg/kg), and (**D**)—(-)-Fenchone (150 mg/kg). Results are expressed as the median (minimum–maximum) of the parameters analyzed (n = 5, three sessions per animal). The Kruskal–Wallis test and Dunn’s posterior test were performed using the GraphPad 5.0 software. ** *p* < 0.01, *** *p* < 0.001 compared to the control group (5% tween 80).

**Figure 9 pharmaceuticals-17-00641-f009:**
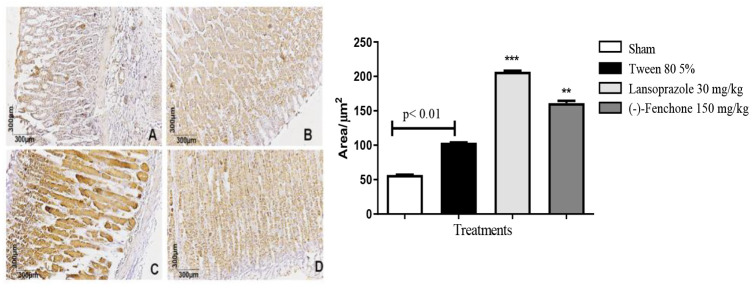
Photomicrographs with immunohistochemical marking for TGF-β in stomach samples from rats: (**A**) sham, (**B**) 5% Tween 80 (10 mL/kg), (**C**) Lansoprazole (30 mg/kg), and (**D**) (-)-Fenchone (150 mg/kg). Results are expressed as the median (minimum–maximum) of the parameters analyzed (n = 5, three sessions per animal). The Kruskal–Wallis test and Dunn’s posterior test were performed using the GraphPad 5.0 software. ** *p* < 0.01, *** *p* < 0.001 compared to the control group (5% tween 80).

**Table 1 pharmaceuticals-17-00641-t001:** Effect of oral administration of (-)-fenchone for 14 days on water, feed, weight evolution, and organ index of rats in the acetic acid-induced ulcer model.

Parameters	Treatments			
	Sham	5% Tween 80	Lansoprazole (30 mg/kg)	(-)-Fenchone (150 mg/kg)
Water intake (mL)	248.2 ± 15.7	207.6 ± 21.7 ^###^	223.8 ± 12.8 *	243.9 ±11.3 ***
Feed intake (g)	182.7 ± 9.5	150.7 ± 5.4 ^###^	165.6 ± 24.2 *	193.0 ± 14.8 ***
Weight evolution (g)	52.3 ± 3.9	35.6 ± 2.6 ^###^	54.8 ± 3.6 ***	42,3 ± 2.1 **
Organ index				
Liver	34.9 ± 1.8	33.5 ± 2.2	33.6 ± 2.6	33.1 ± 1.7
Heart	4.3 ± 0.1	4.1 ± 0.3	4.2 ± 0.2	4.3 ± 0.5
Kidneys	8.2 ± 0.2	8.5 ± 0.6	4.8 ± 0.2	8.3 ± 0.6
Spleen	2.8 ± 0.2	2.7 ± 0.1	3.0 ± 0.1	2.9 ± 0.2

Values are expressed as mean ± SD (n = 6–8). One-way ANOVA revealed significant differences: F (3.52) = 19.61, followed by Dunnett’s or Tukey’s post hoc tests. * *p* < 0.05, ** *p* < 0.01 and *** *p* < 0.001 compared to the 5% Tween 80 group; ### *p* < 0.001 and compared to the sham group. Organ evaluation values are expressed as the organ index, calculated by dividing organ weight (mg) by animal weight (g).

**Table 2 pharmaceuticals-17-00641-t002:** Hematological and biochemical parameters of rats treated for 14 days with lansoprazole or (-)-fenchone in the acetic acid-induced ulcer model.

Parameters	Treatments			
	Sham	5% Tween 80	Lansoprazole (30 mg/kg)	(-)-Fenchone (150 mg/kg)
**Hematological**				
Red blood cells (10^6^/mm^3^)	7.7 ± 0.4	7.3 ± 0.4	7.1 ± 0.4	7.6 ± 0.4
Hemoglobin (g/dL)	14.8 ± 0.4	14.3 ± 1.0	14.0 ± 1.0	14.9 ± 1.2
Hematocrit (%)	39.7± 1.5	39.6 ± 2.9	38.3 ± 2.6	40.9 ± 3.8
MCV (μ^3^)	52.1 ± 2.0	55.4 ± 2.1	52.3 ± 3.9	52.4 ± 3.0
HCM (μg)	19.5 ± 0.7	19.9 ± 0.7	19.1 ± 1.3	19.1 ± 1.0
CHCM (%)	36.6 ± 1.0	36.0 ± 0.8	36.2 ± 1.1	36.7 ± 0.4
Leukocytes (10^3^/mm^3^)	8.2 ± 2.9	9.3 ± 4.2	11.6 ± 5.0	10.7 ± 4.6
Neutrophils (%)	6.1 ± 2.8	8.3 ± 1.5	8.8 ± 3.4	8.9 ± 3.7
Lymphocytes (%)	78.9 ± 5.5	77.1 ± 4.9	75.0 ± 5.9	75.0 ± 6.1
Monocytes (%)	14.8 ± 1.6	13.1 ± 2.1	13.3 ± 2.1	14.9 ± 1.6
**Biochemicals**				
Glucose (mg/dL)	150.7 ± 15.5	147.0 ± 15.3	165.7 ± 23.1	163.6 ± 18.2
Cholesterol (mg/dL)	67.2 ± 6.1	75.0 ± 6.3	71.1 ± 3.9	72.4 ± 6.6
Urea (mg/dL)	47.8 ± 5.6	47.0 ± 4.0	49.1 ± 5.1	49.5 ± 5.5
Creatinine (mg/dL)	0.6 ± 0.0	0.5 ± 0.0	0.5 ± 0.0	0.5 ± 0.0
FAL (mg/dL)	269.2 ± 17.4	313.2± 44.3	302.1 ± 35.9	315.3 ± 41.7
AST (U/I)	138.8 ± 6.3	124.2 ± 10.8	129.0 ± 12.8	131.4 ± 8.2
ALT (U/I)	58.7 ± 7.4	56.3 ± 13.8	54.6 ± 11.0	57.5 ± 8.1

Values are expressed as mean ± standard deviation (n = 6–8). One-way ANOVA: F (3.25) = 2.050, followed by Dunnett’s or Tukey’s post hoc test.

## Data Availability

The data presented in this study are available in this article.

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
