# Peer review of "(-)-Fenchone Prevents Cysteamine-Induced Duodenal Ulcers and Accelerates Healing Promoting Re-Epithelialization of Gastric Ulcers in Rats via Antioxidant and Immunomodulatory Mechanisms"

_pharmaceuticals, 2024, doi:10.3390/ph17050641_

Round 1
Reviewer 1 Report
Comments and Suggestions for Authors
The manuscript written by authors are well explained.
The Introduction part of manuscript is too brief. Authors can add up some relevant information.
*Correct the italic style of ‘in vitro and ex vivo’ in all the text.
Language: At a few instances, the authors failed to comply with the international language standards, the same should be improved upon.
The figure legends and Tables caption need improvement. All legends and captions should have enough description for a reader to understand the figure/table without having to refer back to the main text of the manuscript.
Authors should at the minimum proof read the entire manuscript for typographical errors and fix all grammatical errors.
Value and unit should be separated by a space e.g. 20 ± 2 g (except for % and °C “degrees Celsius”).
Please correct this……..followed by the Dunnet.and Tukey post-test.
Several unnecessary abbreviations have been used; Author could use the abbreviated forms of words which are repeatedly used in the manuscript.
Comments on the Quality of English LanguageLanguage: At a few instances, the authors failed to comply with the international language standards, the same should be improved upon.
Author Response
Dear Reviewer 1,
Corrections were made throughout the manuscript as requested. Some information has been added to the introduction, typographical and grammatical errors have been corrected. All suggested changes were accepted, aiming to improve the quality and accuracy of the content presented as well as the English language of the text where fully revised.

Reviewer 2 Report
Comments and Suggestions for Authors Dear authors,After reviewing the following manuscript entitled "(-)-Fenchone prevents cysteamine-induced duodenal ulcres and accelerates healing promoting re-epithelization of gastric ulcers in rats via antioxidant and immunomodulatory mechanisms” (Pharmaceuticals-2926979), I sent the following comments and observations that the authors should attend to before its publication in this journal.
I appreciate the work of the authors, but please resolve the following data: 1. I recommend that the names of the plants used in the study be written correctly throughout the manuscript (e.g. Phoeniculum vulgare Mill.). 2. Pay attention to mentioning the numbers corresponding to the references if they follow each other in a numerical string (e.g. 1,2,3 is incorrect; 1-3 is correct) 3. The figures should be organized in the text so that it is easier to follow (e.g. in Figure 6 A and B on one row and C and D on another row). 4. There is no information from the manuscript about the method of preparation of the medicinal plants used, namely their origin, the method of harvesting, the preparation of the extracts, the doses administered to rats, the time of administration. 5. The conclusions are completely missing. 6. What is their applicability and importance from a pharmaceutical point of view? 7. The bibliography is not written in accordance with the requirements of the journal. Journal Articles: 1. Author 1, A.B.; Author 2, C.D. Title of the article. Abbreviated Journal Name Year, Volume, page range.8. I recommend the authors to remove the references below the year 2013, especially those from 1995 (ref. 54); 1990 (ref. 55); 1984 (ref. 56); 1988 (ref. 57); 1983 (ref. 58).
Author Response
Dear Reviewer 2,
Corrections were made throughout the manuscript as requested. Typographical and grammatical errors have been corrected.
As described in item 2.2, the substance under study (-)-Fenchone was purchased from SIGMA Chemical Co, U.S.A. (≥ 98% purity). The conclusions topic has been added to the manuscript.
Regarding applicability and importance from a pharmaceutical point of view: Based on the results obtained, fenchone demonstrated promising pharmacological properties that can have diverse applications and importance from a pharmaceutical point of view. Some considerations include: Therapeutic potential for gastrointestinal ulcers: Fenchona's ability to promote the healing of gastric and duodenal ulcers suggests that it can be explored in the development of medicines for the treatment of these conditions. Reduced toxicity: The low toxicity observed in repeated doses over a prolonged period is a crucial aspect of the safety of any pharmaceutical compound. Fenchona demonstrated this characteristic, which could make it a viable option for clinical use. Antioxidant and immunomodulatory mechanisms: Fenchona's ability to modulate the levels of antioxidants and pro-inflammatory and immunoregulatory cytokines suggests that it may be useful in the development of medications intended to treat inflammatory and oxidative stress-related conditions.
As for references, those prior to 2013 refer to articles that cite the methodologies used in the study, therefore, they cannot be removed. All suggested changes were accepted, aiming to improve the quality and accuracy of the content presented as well as the English language of the text where fully revised.

Reviewer 3 Report
Comments and Suggestions for Authors
1. The motivation for the study is not clear. In general, the authors write that it was previously shown that (-)-Fenchone did not perform badly as an antioxidant, etc., probably (-)-Fenchone will save animals from ulcers caused by cysteamine. Very strange! I suggest that authors clearly articulate the motivation for the study in the Introduction section of the manuscript.
2. The model of cysteamine-induced duodenal ulcer is not clear. What processes does such a model allow to reproduce under experimental conditions? Why are we talking only about the duodenum?
3. Cysteamine is known to cause stomach/intestinal disorders and sometimes central nervous system disorders. However, almost all mercaptan derivatives have this effect. Why do the authors use cysteamine?
4. Did the animals smell after working with cysteamine? Did this affect the behavior of the animals? Their physiological status?
5. Lines 418-419: "After 1 h from pretreatment, cysteamine hydrochloride (300 mg/kg) was given twice orally (5 ml/kg) in a 4-h interval induction." At what concentration was cysteamine given: 300 mg/kg or 5 ml/kg? How long did it take to give the treatment? It is extremely difficult to understand what is happening in an experiment! I suggest that the authors write clearly and with illustrations what happens in the experiment? Each step should be labeled with concentration, time of administration, route of administration, etc.
6. Cysteamine is poorly soluble in water. Did the authors use a water-alcohol mixture? What did you do to dissolve cysteamine? Or did they inject the animals with a colloidal solution of cysteamine?
7. How were the authors able to ensure such reproducibility of the damage area? In 5-7 experiments the area differs by several mm2. It seems to me that the authors need to describe the methodological technique used in extreme detail!
8. With statistical research there is confusion throughout the text. For example, graph 1. ***p<0.001, compared with the negative control group (0.9% saline solution) - the 0.9% saline solution group is not on the graph! What did the authors mean? # p<0.001, compared to the lansoprazole group (30 mg/kg) - the first column differs from the lansoprazole group by the maximum value, but the authors do not mark it. Moreover, further there are several columns that are almost no different from the lansoprazole group, all columns are marked.
9. The authors measure interleukins and TNF. It is obvious that with strong inflammation their concentrations change significantly, and with a decrease in inflammation the changes are smaller. The same goes for markers of oxidative stress. The results obtained by the authors in experiments do not in any way support the authors' conclusions that (-)-Fenchone acts through antioxidant and immunomodulatory mechanisms.
10. It should be noted that half of the work was carried out on a model of intestinal damage using acetic acid. How do the data obtained with this acid compare with the data obtained with cysteamine?
I cannot recommend the manuscript for publication.
Little things
In Figure 5, the ordinate axis is marked with strange units area/Zm2.
When listing reagents, we constantly encounter ,, .,. etc.
Author Response
According to the editor's instructions, reviewer 3's questions were disregarded.
Round 2
Reviewer 2 Report
Comments and Suggestions for Authors
I think it can be accepted for publication.
Reviewer 3 Report
Comments and Suggestions for Authors
I did not receive replies to any of my comments, the authors simply ignored them